# 3D Simulations of Freezing Characteristics of Double-Droplet Impact on Cold Surfaces with Different Wettability

**DOI:** 10.3390/e24111650

**Published:** 2022-11-14

**Authors:** Anjie Hu, Qiaowei Yuan, Kaiyue Guo, Zhenyu Wang, Dong Liu

**Affiliations:** School of Civil Engineering and Architecture, Southwest University of Science and Technology, Mianyang 621010, China

**Keywords:** double-droplet impact, icing, VOF model, superhydrophobic

## Abstract

In this work, the freezing characteristics of double-droplet impact on three typical wettability surfaces were investigated by coupling the solidification and melting VOF models. Different temperature conditions were adopted to study the influence of icing speed on droplet behavior. Simulation results show that the motion of the double-droplet impact is consistent with that of a single droplet in the early spreading stage but behaves differently in the retraction stage. The wetting area evolution during the impact-freezing process shows different tendency for hydrophilic and hydrophobic surfaces: Compared with single droplets, double droplets have a smaller wetting area factor on hydrophilic surfaces but a larger one on superhydrophobic surfaces. In addition, three typical impact results are observed for the double-droplet impact on a superhydrophobic cold surface: full rebound, adhesive avulsion, and full adhesion, which reflects the interaction of droplet merging and solidification during the impact freezing of the double droplet. These findings may deepen our understanding of the mechanism of impact freezing on a cold surface, it provides reference for the associated applications and technologies in icing/anti-icing.

## 1. Introduction

The impact–freezing process of droplets impacting cold surfaces is common in engineering applications. For example, when an aircraft passes through a cumulonimbus cloud at high altitudes at low temperatures, the supercooled droplets in the cloud will impact and freeze on the cold wing surface. This disrupts the aerodynamic properties of the aircraft and, in severe cases, may even lead to the loss of control [1]. Additionally, the icing problem is a threat to the air-conditioning system [2,3,4,5], wind turbines [6,7,8], and power transmission systems [9,10].

Inspired by the “lotus leaf effect” in nature [11], researchers have created a biomimetic superhydrophobic surface with low adhesion and self-cleaning properties and have obtained a solid theoretical foundation by investigating the impact dynamics of droplets on room-temperature surfaces through experimental studies and numerical simulations [12,13]. However, for impact dynamics on cold surfaces, the underlying mechanism is complex and unclear due to the interaction between droplet impact and freezing; therefore, the impact–freezing mechanism of droplets on superhydrophobic cold surfaces needs further investigation.

In recent years, the impact–freezing process of droplets on cold surfaces has attracted increasing scholarly attention. Bodaghkhani et al. [14] investigated the total freezing time of droplets on surfaces with different wettability with horizontal and inclined orientations and showed that lower surface temperature, smaller static contact angle, and higher inclination can accelerate the freezing of droplets. Moghtadernejad et al. [15] studied high-speed droplet impact on cold surfaces with different wettability. It was found that on a hydrophilic substrate, the droplets form a rivulet, which then freezes on the cold plate. In contrast, there is no rivulet formation on the superhydrophobic surface. Li et al. [16] experimentally investigated the impact process of supercooled droplets on hydrophilic and hydrophobic cold surfaces. It was found that when a supercooled droplet impacts a cold surface, solidification greatly hinders the droplet’s rebound on the superhydrophobic cold surface. In an experiment to visualize the freezing mass of a supercooled droplet impinging on a cylindrical cold surface, Yang et al. [17] found that air temperature, surface temperature, and supercooled droplet temperature are all important components affecting droplet solidification. Jin et al. [18] investigated the impact and freezing process of droplets on different cold cylindrical surfaces by experimental methods, and they found that the spread factor of droplets at low surface temperatures was greater than that of droplets at room temperature. The effect of surface temperature on the impact and freezing processes of droplets on cold stainless steel surfaces was investigated experimentally and numerically by Yao et al. [19]. The surface temperature had a small effect on the spreading phase but a significant effect on the shrinkage rate and the final equilibrium state. Xu et al. [20] investigated the impact–freezing behavior of droplets impacting cold solid surfaces at different Weber numbers by experimental methods and concluded that a higher impact velocity significantly improved the retraction of the droplet, but this effect was highly reduced on a cold solid surface at low temperatures.

In addition to experimental studies, to better observe the phase shift motion during the impact icing process, researchers have begun to investigate the impact–freezing properties of droplets using numerical models as a complement to experimental data. Blake et al. [21] developed a numerical model of the impact–freezing process of a supercooled droplet on a cold surface based on the VOF model and the solidification/melt model, and successfully predicted several different impact responses of the droplet. Tembely et al. [22] performed a numerical simulation of the droplet impact, diffusion, and freezing processes after impacting a cold superhydrophobic surface and found that the solidification time varied exponentially with the maximum expansion diameter of the droplet. Chang et al. [23] applied this method to solve the solidification problem and analyzed the evolution of the velocity, temperature, and heat flow distribution during the phase change of a supercooled droplet. Zhang et al. [24] also investigated the properties of subcooled droplets impacting a cold surface through experiments and simulations. By coupling the VOF model and the solidification/melt phase change model, the results obtained were in good agreement with the corresponding experimental results, with a maximum deviation of 11.3% for the stable spreading factor.

These studies have revealed the deeper mechanisms of the droplet icing process. However, they focus mostly on individual droplets, which cannot reflect the icing situation when multiple droplets hit a superhydrophobic surface at the same time. As droplets combine with each other to form larger droplets, the strong contact between adjacent droplets during diffusion and solidification may make the collision dynamics extremely complex, which is a key difference between multi-droplet collisions and single-droplet collisions [25,26,27]. Recently, several researchers [28,29,30] have numerically studied the impact behavior of multiple droplets on hydrophilic and hydrophobic surfaces. These studies showed that droplet-to-droplet aggregation dynamics significantly altered the droplet spreading and shrinking behavior, leading to significantly different results from single-droplet collisions. Wang et al. [31] studied the rebound dynamics of double droplets impacting a flat superhydrophobic substrate simultaneously and found three rebound states based on the center distance between the double droplets: a fully coalescent rebound (CCR) state, a partially coalescent rebound (PCR) state, and a non-coalescent rebound (NCR) state.

From the above literature, it can be seen that, in the droplet impact-freezing studies, most of the research is carried out mainly on the single droplet impact-freezing process, while in real life, impact freezing of multiple droplets is more common. Although researchers have conducted some work studying the impact behavior of double droplets, the impact-freezing characteristics of double droplets are barely investigated. In order to investigate the motion and icing of multiple droplets on a solid surface, the numerical simulation based on the commercial CFD software ANSYS Fluent [32] is adopted to investigate the impact–freezing characteristics of double droplets impacting on a cold surface with different wettability at the same time. The impact and solidification processes of these droplets were investigated by coupling the VOF model with the solidification/melt model [24,33]. The effect of different contact angles on the freezing process were investigated for the same spacing conditions. The evolution of the wetting area factor and morphology of the double droplet during impact freezing was obtained and used to study the properties of the aggregation and freezing processes of double droplets impacting on a cold surface at the same time.

## 2. Numerical Model

### 2.1. Multiphase Model

To obtain the motion and deformation of the gas-liquid interface during the impact freezing of droplets on cold surfaces, the current numerical model uses the volume of fluid (VOF) multiphase model [33] in the ANSYS fluent framework. The VOF method is based on the Euler method to track the gas-liquid free phase surface. In this model, the volume fraction of each computational grid cell is given as
(1)$$\alpha_{j}=\frac{\mathrm{volume}\;\mathrm{of}\;\mathrm{jth}\;\mathrm{phase}}{\mathrm{cell}\;\mathrm{volume}},$$
where $\sum \alpha_{j}=1$. In this simulation, the water–ice mixture, ice, and liquid water are all treated as one liquid phase, and the air is treated as the second phase. The mass transition equation of two-phase flow can be represented as:(2)$$\frac{\partial }{\partial t}(\alpha_{j}\rho_{j})+\nabla \left({\;\alpha }_{j}\rho_{j}\vec{u}\right)=0,\;j=1,\;2.\;$$

The corresponding momentum conservation equation is given by:(3)$$\frac{\partial }{\partial t}(\rho \vec{u})+\nabla (\rho \vec{u}\vec{u})=-\nabla p+\nabla \mu (\nabla \vec{u}+{\left(\nabla \vec{u}\right)}^{T})+\rho \vec{g\;}+\vec{F_{\mathrm{st}}}+\vec{S_{M}},$$
where ρ is the total fluid density, and μ is the average viscosity in each cell, which is given by
(4)$$\rho \;=\sum_{j}\left(\alpha_{j}\rho_{j}\right),\mu =\sum_{j}\left(\alpha_{j}\mu_{j}\right),\;j\;=\;1,\;2.\;$$

$\vec{F_{\mathrm{st}}}$ is the volume surface tension force acting on the gas–liquid interface. $\vec{S_{M}}$ is the momentum source term introduced by the freezing process. In this work, the continuum surface force (CSF) model proposed by Brackbill et al., is adopted to calculate the surface tension force:(5)$$\vec{F_{\mathrm{st}}}\;=\;\sigma \frac{\rho \kappa \nabla \alpha }{0.5\left(\rho_{1}{+\rho }_{2}\right)},\;$$
where σ is the surface tension coefficient, and κ is the curvature of the interface, which is given by
(6)$$\kappa \;=\;{\left.\nabla \cdot \frac{\nabla \alpha }{\left|\nabla \alpha \right|}\right|}_{\alpha =0.5}.\;$$

The contact angle at the wall was set by calculating the free interfacial normal vector in the control body at the wall, which changes the interfacial curvature and surface tension source terms; the normal vector is given by
(7)$$\vec{n}{\;=\;\vec{n}}_{w}{\;\cos\theta }_{w}{+\vec{t}}_{w}{\;\sin\theta }_{w},\;$$
where ${\vec{n}}_{w}$ and ${\vec{t}}_{w}$ are the unit vectors normal and tangential to the wall, respectively, and $\theta_{w}$ is the prescribed contact angle.

### 2.2. Solidification/Melting Model

Experimental and numerical studies by Zhang et al. [24] show that after a super-cooled droplet touches a surface, the supercooled surface takes away the heat from the droplet, causing it to gradually solidify. The freezing of a supercooled droplet on a cold surface consists of two processes: nucleation and recalescence. As the recalescence phase is fast and has a complex triggering mechanism, the supercooled impacting water droplet with an initial velocity and an initial temperature is assumed to complete the nucleation-recalescence stage upon touching the cold surface or in a very short time [34,35].

In this work, the enthalpy-porosity phase change model [24] is adopted to simulate the solidification-melting phase change process inside the droplet, neglecting the recalescence phase, and the energy equation is given by
(8)$$\frac{\partial }{\partial t}(\rho h)+\nabla (\rho \vec{u}h)\;=\;\nabla \left(\lambda \nabla T\right),\;$$
where λ is the thermal conductivity, the enthalpy value h is the sum of latent enthalpy $h_{\mathrm{ls}}$, and sensible enthalpy is $h_{\mathrm{se}}$:(9)$${h=h}_{\mathrm{se}}{+h}_{\mathrm{ls}}.\;$$

The sensible enthalpy $h_{\mathrm{se}}$ is given by
(10)$$h_{\mathrm{se}}{\;=h\;}_{\mathrm{ref}}+\int_{T_{\mathrm{ref}}}^{T}c_{p}\mathrm{dT},\;$$
where $h_{\mathrm{ref}}$ and $T_{\mathrm{ref}}$ are the reference enthalpy and reference temperature, respectively, and $c_{p}$ is the thermal capacity. The latent enthalpy is determined by the liquid water fraction in the droplet during the solidification/melting process:(11)$$h_{\mathrm{ls}}\;=\;L\gamma ,\;$$
where L is the latent heat, and γ is the liquid water fraction which is assumed to be dependent on the temperature:(12)$$\gamma =\left\{\begin{array}{c} 0\\ \frac{{T\;-\;T}_{\mathrm{solid}}}{T_{\mathrm{liquid}}-T_{\mathrm{solid}}}\\ 1 \end{array}\right.\begin{array}{c} {\;T\;<\;T}_{\mathrm{solid}}\\ \;T_{\mathrm{solid}}\;\leq \;T\;\leq {\;T}_{\mathrm{liquid}}\\ {\;T\;>\;T}_{\mathrm{liquid}} \end{array},\;$$
where $T_{\mathrm{solid}}$ and $T_{\mathrm{liquid}}$ are the critical temperatures when the droplet completely solidified/melted. The interval of $\left[T_{\mathrm{solid}},\;T_{\mathrm{liquid}}\right]$ is always set small to keep the interface of the ice and water shape.

In previous studies, the enthalpy-porosity method was always adopted to simulate the ice-water mixing area. In this method, the ice-water mixing area is treated as a porous area, and the momentum source term $\vec{S_{M}}$ in Equation (3) is given as
(13)$$\vec{S_{M}}=\frac{{\left(1-\gamma \right)}^{2}}{\gamma^{3}+\varepsilon }C_{\mathrm{mush}}\vec{v}.\;$$

In this equation, ε is the minimum value to avoid a zero denominator (ε = 0.001); C_mush_ is the viscosity coefficient, which is related to the shape of the porous medium. For the value of C_mush_, a range of values between 10–4 and 10–7 is used in most cases [36], which is related mainly to the surface temperature and wettability and has no significant correlation with the physical properties of the droplet and the impact velocity, and in the present work, the value of C_mush_ was in agreement with the literature [24].

This work used the commercial CFD software ANSYS Fluent to perform numerical simulations. The pressure-based solver was used to calculate the transient impact, and the PISO (Pressure Implicit Splitting Operator) format was adopted to couple pressure and velocity. To precisely capture the interface of the droplet, the geometric reconstruction format was adopted. To save computational resources and ensure the accuracy of the iterative results, adaptive time steps were used to control the Courant number to be always less than 0.2, and the time discretization format was a first-order implicit format with 20 iterations in each time step [37].

### 2.3. Mesh Independency Validation

In order to study the morphology of evolution and phase change heat transfer of double-droplet impacting on cold surfaces with different wettability on the same horizontal plane, a three-dimensional rectangular region with a length, width, and height of 20 mm × 7.5 mm × 7.5 mm was designed. The calculation model and boundary conditions used in this work are shown in Figure 1. The bottom of the model was a constant temperature non-slip wall boundary. The upper surface, left surface, right surface, and front surface were set as pressure outlet boundary conditions. To save the simulation resources, the rear surface was set as a symmetric surface. The property settings of the phases in the simulation were consistent with those reported in the literature [24]. In order to quantitatively study the morphology evolution of droplet impact on a solid surface, a spreading factor [13] was set for evaluating the grid irrelevance in this work as shown in Figure 2, which is defined as:(14)$$\beta \;=\;\frac{D}{D_{0}}$$
where D_0_ is the initial diameter of the droplets (mm), and D is the spreading diameter of the droplets (mm).

In this work, structured meshes were used to divide the entire rectangle, and three different meshing methods were selected to predict by comparing the spreading factors of room-temperature droplets hitting the room-temperature surface. The final result is shown in Figure 3. It can be seen from the figure that the simulation results with mesh sizes of 3.6 million and 9.7 million were very close. Considering the solution accuracy and computational efficiency, this work finally used the hexahedral mesh with the size of 3.6 million.

### 2.4. Numerical Model Validation

In this work, the accuracy of the solidification/melting model was verified using experimental and simulation data from the literature [24]. The single droplet impacting and icing on cold surfaces were simulated using the model and the simulation results were compared with reference data. The simulation parameters were given as follows: $D_{0}$ = 2.84 mm, $T_{0}$ = 0.1 °C, $T_{a}$ = −5 °C, $T_{s}$ = −30 °C, $V_{0}$ = 0.7 m/s, and $\theta_{w}$ = 160°, where $D_{0}$ is the diameter of the droplet, T_0_ is the initial temperature of the droplet, $T_{a}$ and $T_{s}$ are the ambient and solid surface temperatures, respectively, and $V_{0}$ is the impact velocity. Figure 1 shows a comparison of the droplet shape evolution between the simulated and experimental results. As shown in Figure 4, the simulation data calculated from the coupled VOF model and the solidification/melting model agreed well with Zhang’s experiments, indicating that the coupled model could accurately predict the dynamics and icing characteristics of droplets impacting on cold surfaces.

## 3. Results and Discussion

In this work, the impact of double droplets on the cold surface was simulated under different conditions of surface temperature and contact angle. Three typical temperature conditions were adopted: room-temperature droplets impacting the room-temperature surface, room-temperature droplets impacting the cold surface, and supercooled droplets impacting the cold surface. To show the influence of the droplet interaction on the impact, the single-droplet impact was also simulated and compared in the simulation. Table 1 shows the combination of parameters simulated in this work, including contact angle, air temperature, droplet temperature, droplet velocity, and surface temperature, where the droplet spacing was 3.5 mm.

### 3.1. Morphology Evolution of the Droplets

To illustrate the influence of the interaction between droplet on the impact, the morphology evolution of the droplets during the impact is presented in this work. The simulations of three temperature conditions are shown in Figure 5, Figure 6 and Figure 7.

Figure 5 shows the morphology evolution of room-temperature double droplets impacting different wettable room-temperature surfaces simultaneously. It can be seen from Figure 5a that these droplets spread horizontally on the surface, and then shrank after reaching the maximum spreading area. The diffusion process of the droplets on the surface lasted significantly longer than the shrinkage stage. Due to the influence of surface tension and friction loss, when the droplet reached the maximum spreading area, the liquid film on both sides of the central liquid ridge gradually diffused and diluted until tearing, formatting a central liquid ridge and double marginal droplets at approximately 30 ms. Figure 5b shows the morphological evolution of room-temperature double droplets impacting on the room-temperature hydrophobic surface. With the increase of contact angle, the droplet had greater forward resistance, and the spreading behavior was hindered. In the spread stage, the droplets spread around the impact center, forming a double circle shape, until the droplet edges contacted and converged to form liquid ridges, which was almost the same as the morphological evolution of the single-droplet impact [13,38]. In the contraction stage, the double-droplet impact produced a cohesive force during the merging process. Under the influence of the force, the retraction speed of the droplet became asymmetric, resulting in the elliptical contraction shape of the double droplet. When the surface contact angle reached 160° (Figure 5c), the droplet shape was close to the hydrophobic surface case. The friction loss between the droplet and the surface was further reduced, and the remaining kinetic energy of the droplet was further increased. Finally, the droplet began to bounce off the surface completely at approximately 20 ms. In previous research of a single-droplet impact on the cold surface with no phase change, the shape of the contact area of the droplet and the solid surface maintained a circle during the spread and contraction process [13,31], which was obviously different from the double-droplet impact case in the present work, showing that the interaction of droplets significantly influences the morphology of droplets during impact.

Since the freezing process of the droplet impacting on the cold surface changes the droplet adhesion on the wall, this paper further analyzed the simulation results of double typical impacts of the normal-temperature droplets and the cold droplets impacting on cold surfaces.

Figure 6 shows the morphological evolution of room-temperature double droplets impacting cold surfaces with different wettability, where the initial droplet temperature and air temperature were both 15 °C, and the surface temperature was −30 °C. As can be seen in Figure 6a, in the early spread stage, the droplets’ behavior was very similar to the normal-temperature case shown in Figure 5a. While in the shrinkage stage, a large heat exchange area occurred between the bottom liquid and the cold surface, leading to ice formation at the bottom of the droplet. As the upper liquid continued to flow, the droplet eventually formed an elliptical disk and rested on the surface. Figure 6b shows the morphology evolution of the room-temperature droplet impacting a cold hydrophobic surface. As can be seen from the figure, in the spread stage, the spreading behavior of the droplets was hindered by increasing the surface contact angle, which was consistent with the normal-temperature surface impact shown in Figure 5b. In the contraction stage, the droplet on the cold hydrophobic surface was blocked in the *z*-direction due to the icing at the bottom and the cold surface, which caused the residual kinetic energy of the unfrozen part of the fluid in the upper region of the droplet to decrease sharply. The retraction height of the droplet was reduced compared with the normal-temperature case. As the contact angle continued to increase to 160°, it can be seen in Figure 6c that the droplet in the spreading phase was also consistent with the spreading process in Figure 5c. As the viscous friction between the droplet and the surface continued to decrease, the remaining kinetic energy increased and retraction accelerated the fluid flow rapidly toward the impact center before becoming completely frozen. The contact area between the droplet and the cold surface reached a minimum at the time of 20 ms. Compared with Figure 5c, these droplets did not bounce off the cold surface after impacting the superhydrophobic surface. These results show that the icing of the droplets significantly reduced the hydrophobic effects and increased the adhesion of the surface.

Figure 7 shows the morphological evolution of supercooled double droplets impacting on the cold surface with different wettability, where the droplet temperature was 0.1 °C, the air temperature was −5 °C, and the surface temperature was −30 °C [24,39]. Compared with Figure 6, it can be seen from this figure that, although the supercooled environment enhanced the heat transfer between droplets and air, it had less effect on the spread stage where the double droplet impacted the cold surface. However, in the contraction stage, the droplets in the supercooled environment produced more ice, making the bottom fluid more susceptible to freezing. At the same time, heat exchange occurred between the outer edge fluid and the cold air to form an ice–water mixture, significantly slowing down the contraction of the droplets. The droplets impacting the superhydrophobic surface adhered to the wall, and the final contact area was obviously larger than the normal-temperature droplet case.

From the above results, we can see that the double-droplet collision morphology changed significantly compared with a single droplet, and the droplet surface changes were more complex due to the interaction of the double droplet. However, the influence of icing is more significant in the spreading stage. The main reason for this phenomenon is that the droplet spreading stage is characterized by a high speed of interfacial movement, a small amount of icing, and a low impact of the additional viscous drag caused by icing; thus, the droplet movement was dominated mainly by the droplet inertial force and was, therefore, largely unaffected by icing. In the retraction stage, as the droplet velocity decreased, the inertial force decreased and the amount of icing increased, the droplet motion was dominated mainly by the adhesive force caused by icing.

### 3.2. Wetting Area Evolution Characteristics

The size of the wetting area during the impact is the key factor that determines the cooling rate and adhesion of the wall. By analyzing the change in the wetting area evolution during droplet impact freezing, one can further understand the surface wettability performance during impact. Unlike the single-droplet impact, the wetting factor defined as the ratio of wetting diameter to droplet initial diameter, which is widely applied, is not suitable for the double-droplet impact, since the shape of the contact area between the droplet and the wall is irregular. To better reflect the wetting area evolution, the wetting area factor is pointed out in this work, which is defined as the ratio of the wetted area of droplets on the solid surface to the sum of the droplet initial cross-sectional areas:(15)$$\beta \;=\;\frac{A_{\mathrm{wet}}}{{n\pi R}^{2}},\;\;$$
where $A_{\mathrm{wet}}$ is the wetting area, n is the number of the droplets, and R is the initial radius of the droplet.

Figure 8 shows the wetting area factor evolution of droplets impacting different wettability surfaces under normal temperature conditions. As can be seen in this figure, since there was no phase change at the wall, the maximum wetting area factor decreased with the increase of the contact angle for both single- and double-droplet impacts. Specifically, for the hydrophilic surfaces, the wetting area factor of single-droplet impact was obviously larger than that of the double-droplet impact. However, in the hydrophobic and superhydrophobic cases, the maximum wetting area factor of the double-droplet impact was almost identical to that of the single-droplet impact, showing that the droplet wetting ability was less affected by the merging of droplets or hydrophobic and superhydrophobic surfaces. To further compare the water-repelling ability of superhydrophobic surface in single-droplet and double-droplet impacts, the recovery coefficient, which is defined as the ratio of the bounce speed to the impact velocity, was compared in Table 2. As shown in this table, the recovery coefficient for a double-droplet impact was much greater than that for a single-droplet impact, indicating that the kinetic energy consumption was reduced in the double-droplet impact case. One possible reason is that the surface tension energy released in the merging process was larger than the extra dissipated kinetic energy due to the vibration.

Figure 9 shows the wetting area factor evolution of room-temperature droplets impacting cold surfaces with different wettability. It can be seen from this figure that the droplets stabilized after a single spreading retraction, and the maximum wetting area factors were almost the same as the normal-temperature droplet case in Figure 8, as well as the times when the wetting area factors reached maximum. These results indicate that icing of droplets had less influence on the droplet spreading stage and more influence during the retraction stage. The influence of droplet interaction on the wetting area factor was also similar to the normal-temperature case: the wetting area factor of double-droplet impact was smaller than that of the single-droplet impact for the hydrophilic surfaces, while in the case of superhydrophobic surfaces, the wetting area factor of double-droplet impact was obviously larger than the single-droplet impact, which shows a higher adhesive force.

Figure 10 shows the wetting area factor evolution of room-temperature droplets impacting cold surfaces with different wettability. It can be found from this figure that the maximum wetting area factors and the influence of the droplet merging on the wetting area factors are consistent with those shown in Figure 9. The vibration of wetting area factors after retraction was further reduced due to the solidification at the bottom of the droplet. The final wetting area factors were largest compared with Figure 8 and Figure 9.

In conclusion, the temperature had little influence on the maximum wetting area factor; however, the final stable wetting area factors increased with the decrease of the temperature. The influence of droplet interaction during the merging process on the wetting area factor varied for surfaces with different wettability: the double-droplet impact showed smaller wetting area factor on the hydrophilic surface while larger on the superhydrophobic surface. This phenomenon can be explained by analyzing the morphological evolution of a subcooled single droplet striking a superhydrophobic cold surface, as shown in Figure 11. It can be seen from this figure that, when a single droplet struck a superhydrophobic cold surface, an air pocket formed at the bottom of the droplet, ultimately reducing the contact area between the liquid and the cold surface. In the double-droplet impact case, the air layer at the bottom escaped due to the vibration in the merging process, increasing the contact area between the droplets and the cold surface. Since the superhydrophobic surface showed the most complex droplet behavior during the impact, in the following discussing, the impact codification phenomenon is further studied.

### 3.3. Heat Transfer and Solidification Characteristics on Superhydrophobic Surface

The evolution of the ice phase on the superhydrophobic surface under different initial temperature conditions has been analyzed in this paper, as shown in Figure 12. It can be seen from this figure that for the normal-temperature droplets, due to the high droplet temperature (15 °C), the freezing of the inner side of the droplet lagged compared with the outer side surface, and the fluid partially broke away from the cold surface before becoming fully frozen and adhering. In the supercooled droplets case, the icing speed was obviously faster than the normal-temperature one, and the solid icing caused the contact area to remain large during the retraction stage, which further increased the icing speed of the droplet. It can also be found that the icing speed of double-droplet impact near the wall was higher than that of single fluid. The reason could be that the air at the bottom during the impact was exhausted due to the vibration caused by droplet merging. These results show that the droplets more easily froze in the double-droplet impact condition.

### 3.4. Influence of the Impact Velocity and Supercooled Degree on the Result of Double-Droplet Impact

To further investigate the anti-icing performance of the superhydrophobic surface, the double-droplet impact on the supercooled superhydrophobic surface is further discussed under the conditions of different impact speed and temperature conditions, as shown in Table 3. It was found that there were three main different morphological states after impacting a cold surface at different temperatures: full rebound, adhesive avulsion, and full adhesion, as shown in Figure 13, where adhesive avulsion represents the result that the merged droplet split during the retraction stage on the combined action of droplet vibration and adhesion of the wall. The partial rebound was not observed in the present study. The reasons could be that, for single-droplet impact, the contact area was symmetrical and the contact line moved evenly in the retraction stage, while for double-droplet impact, the horizontal vibration due to the asymmetric contraction consumed more energy, making it difficult to break from the neck. It can also be seen from the figure that the droplet shape evolutions were consistent at the spread stages, which was similar to the previous studies of a supercooled single droplet impact. In the retraction stage, the droplet shapes were significantly influenced by the supercool degree of the cold surface.

Figure 14 shows the simulation results of different velocity and solid surface temperatures. It can be found in this figure that the droplets were more likely to rebound at low impact velocity and high surface temperature conditions since the contact area was small, and the solidification speed was slow in this case. When the surface temperature was low ($T_{s}$ = −30 °C) and as the solidification speed increased, the droplets were more likely to fully adhere to the solid surface. The adhesive avulsion appeared at the condition of high impact velocity or the middle surface supercooled degree. A high impact velocity can generate strong vibration, which tears the droplet apart during the impact. In the case of $T_{s}$ = −20 °C, the solidification near the wall generated a moderate adhesive force that increased the deformation of the droplet but did not significantly reduce the liquidity of the droplet, which led to the adhesive avulsion at low impact temperature condition. These results show that the impact result was associated with both the supercooled degree of the solid surface and the droplet merging process.

## 4. Conclusions

In this work, the double-droplet impact–freezing phenomena on cold surfaces are investigated by coupling the VOF model with the solidification/thawing model, and the effects of cold surface temperature, ambient temperature, and wettability on the are discussed. Simulation results show that the merging of droplets during the impact can significantly influence the shape of the droplet and enhance the rebound ability of the droplets on the superhydrophobic surface. These results provide a deeper understanding of icing mechanism of multi-droplet impact, which is helpful for developing of large-scale icing model. On the basis of the simulation results, the following conclusions can be made:
(1)In the early spreading stage, the double-droplet impact behaves in the same way as the single-droplet. The influence of temperature conditions has little influence on the droplet dynamics at this stage.(2)The temperature conditions have a significant influence on the retraction stage of the double-droplet impact. The lower the temperatures are, the stronger the adhesion of the wall, and the larger the corresponding wetting area.(3)The wetting area evolution during the impact–freezing process shows different tendency for hydrophilic and hydrophobic surfaces: compared with single droplets, double droplets have a smaller wetting area factor on hydrophilic surfaces while a larger one on superhydrophobic surfaces.(4)Three typical impact results are observed for the double-droplet impact on a superhydrophobic cold surface: full rebound, adhesive avulsion, and full adhesion, which reflects the interaction of droplet merging and solidification during the impact-freezing of the double-droplet.

## Figures and Tables

**Figure 1 entropy-24-01650-f001:**
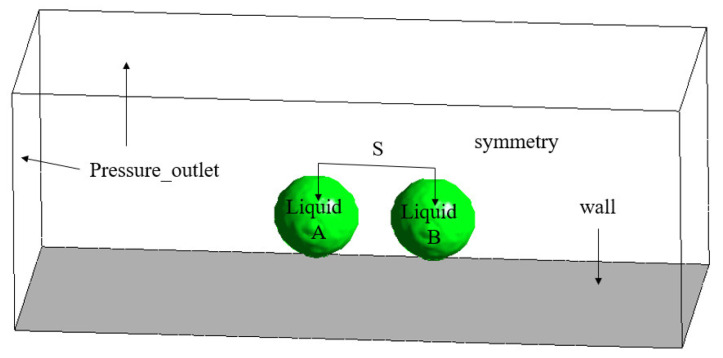
Schematic diagram of physical model and boundary conditions.

**Figure 2 entropy-24-01650-f002:**
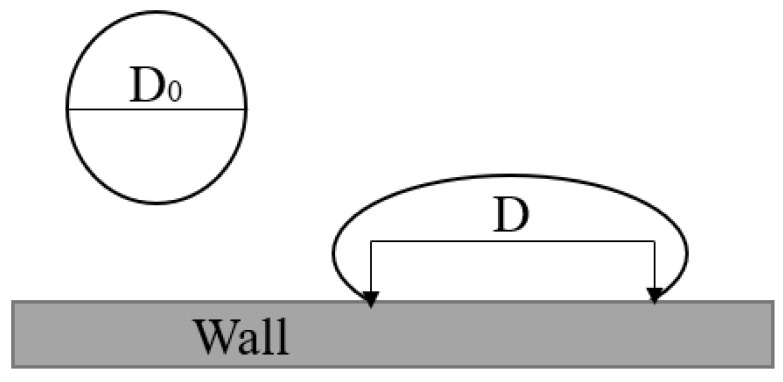
Schematic diagram of droplet shape parameters.

**Figure 3 entropy-24-01650-f003:**
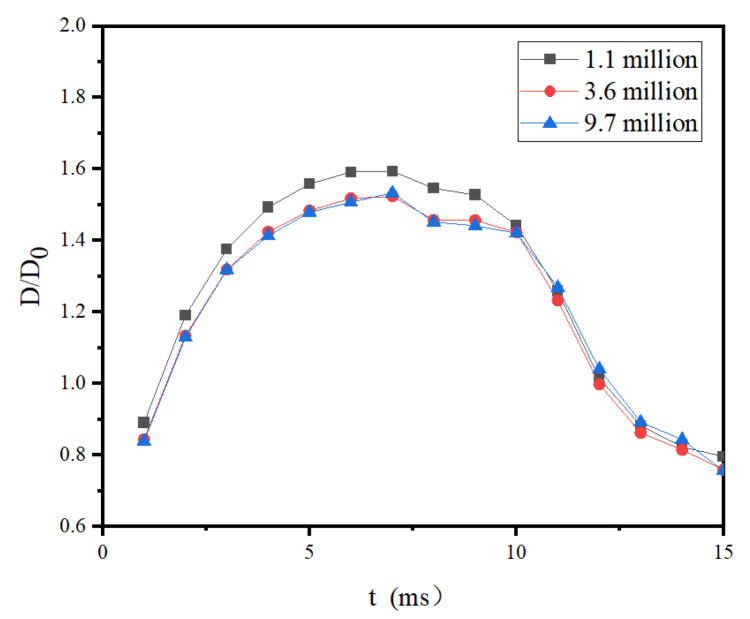
Comparison of spreading factor evolutions with different grid size.

**Figure 4 entropy-24-01650-f004:**
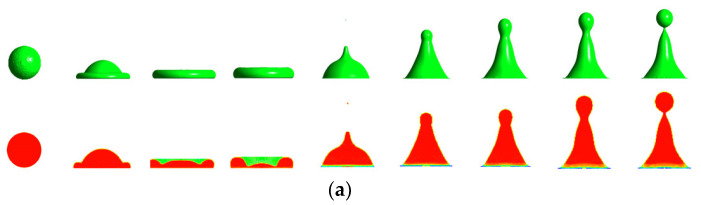
Comparison of simulation and experimental results of droplets impacting on superhydrophobic cold surface (T_a_ = −5 °C, T_s_ = −30 °C): (**a**) droplet shape evolution of the current simulation; and (**b**) comparison of spreading factor over time.

**Figure 5 entropy-24-01650-f005:**
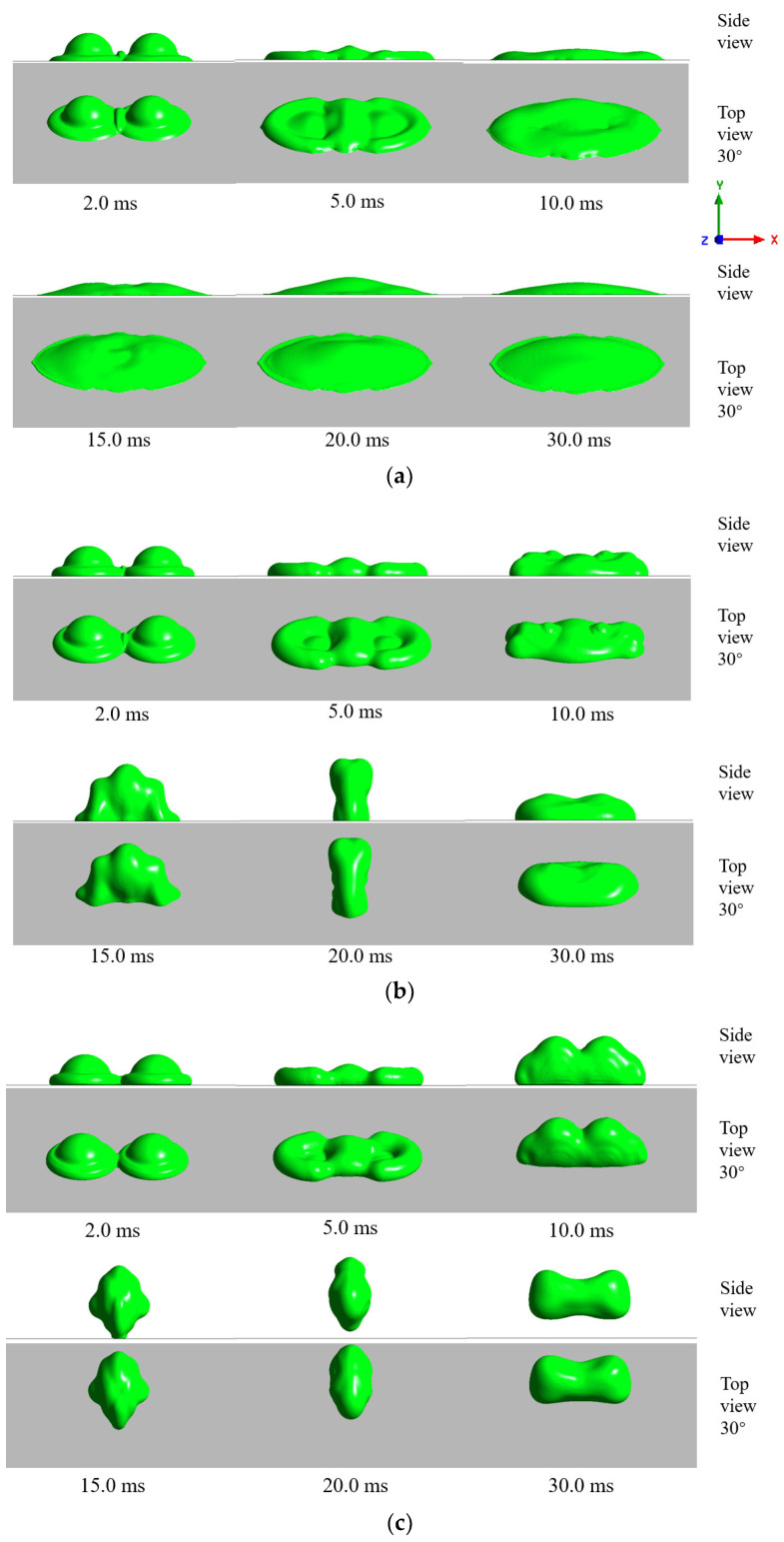
Morphology evolution of room-temperature double droplets simultaneously impacting different wettable room-temperature surfaces: ($T_{a}$ = 15 °C, $T_{0}$ = 15 °C, and $T_{s}$ = 15 °C): (**a**) hydrophilic surface; (**b**) hydrophobic surface; and (**c**) superhydrophobic surface.

**Figure 6 entropy-24-01650-f006:**
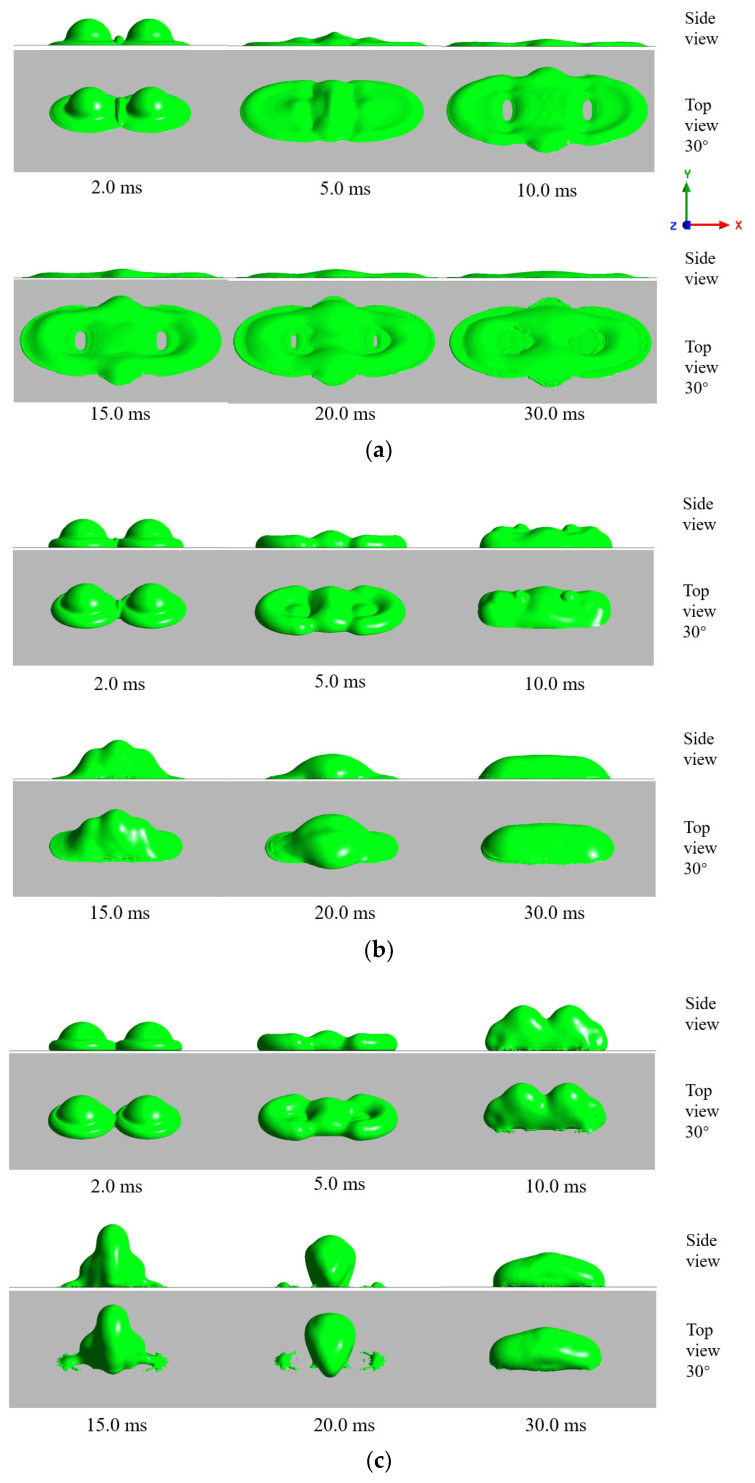
Morphology evolution of room-temperature double droplets simultaneously impacting different wettable cold surfaces: ($T_{a}$ = 15 °C, $T_{0}$ = 15 °C, and $T_{s}$ = −30 °C): (**a**) hydrophilic cold surface; (**b**) hydrophobic cold surface; and (**c**) superhydrophobic cold surface.

**Figure 7 entropy-24-01650-f007:**
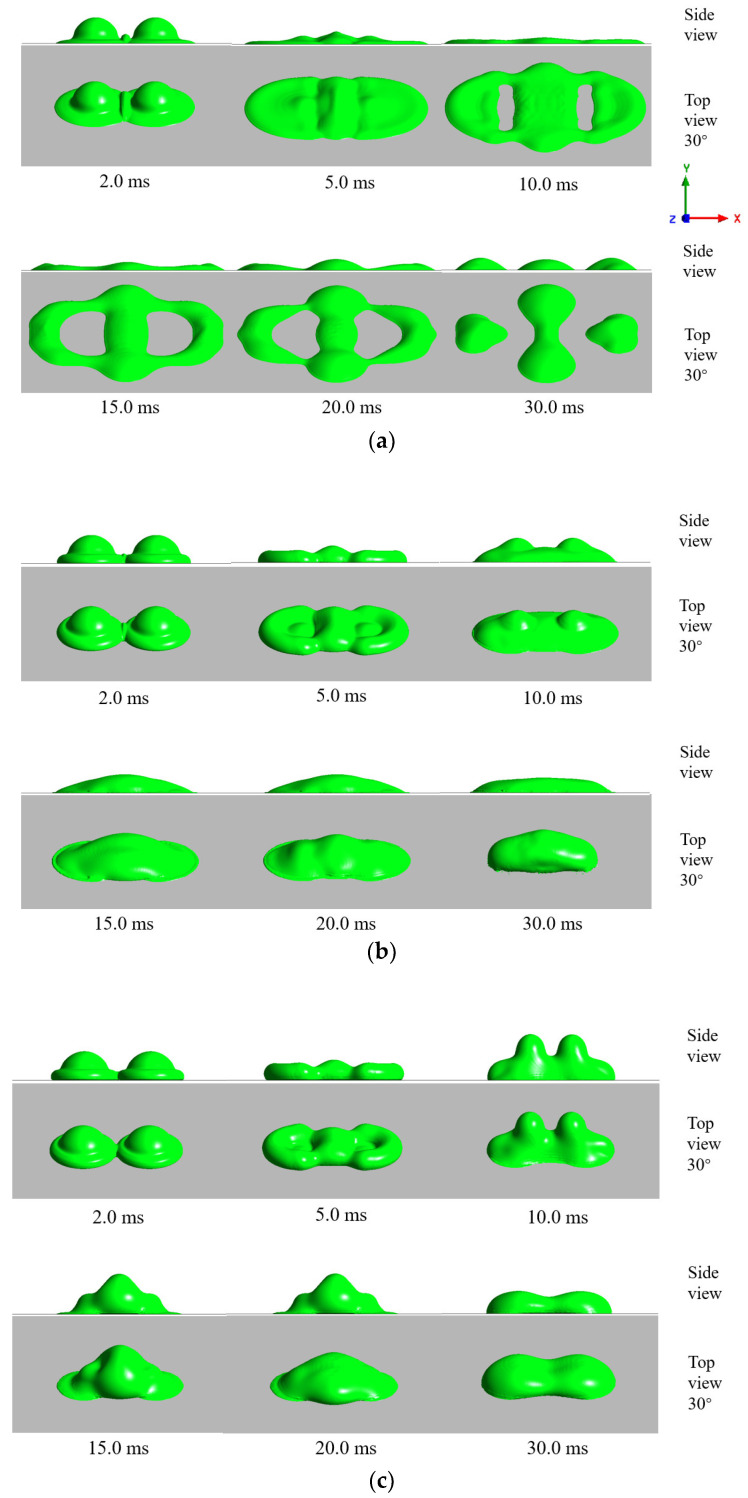
Morphology evolution of supercooled double droplets simultaneously impacting different wettable cold surfaces: ($T_{a}$ = −5 °C, $T_{0}\;$ = 0.1 °C, and $T_{s}\;$ = −30 °C): (**a**) hydrophilic cold surface; (**b**) hydrophobic cold surface; and (**c**) superhydrophobic cold surface.

**Figure 8 entropy-24-01650-f008:**
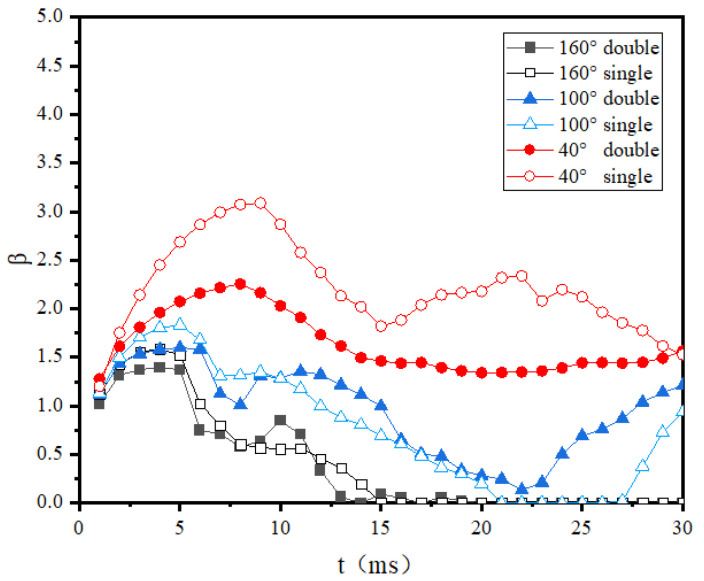
Wetting area factor evolution of room-temperature double droplets impacting room-temperature surfaces with different wettability.

**Figure 9 entropy-24-01650-f009:**
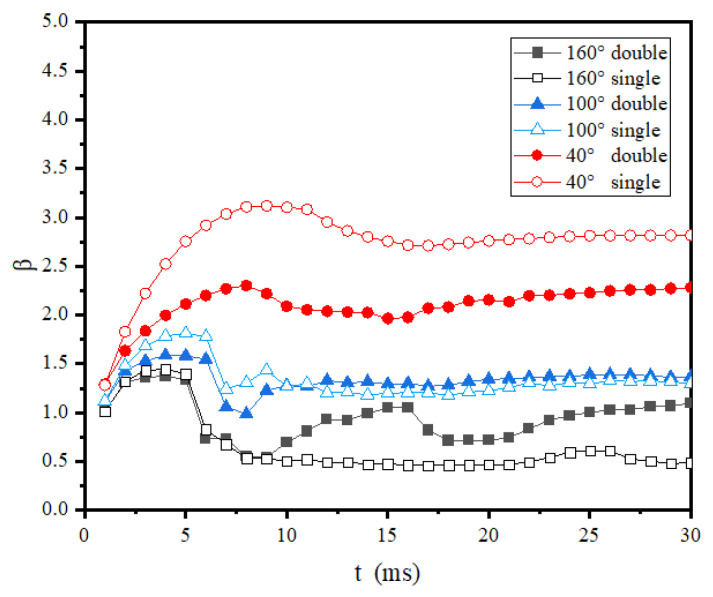
Wetting area factor evolution of room-temperature double droplets impacting cold surfaces with different wettability.

**Figure 10 entropy-24-01650-f010:**
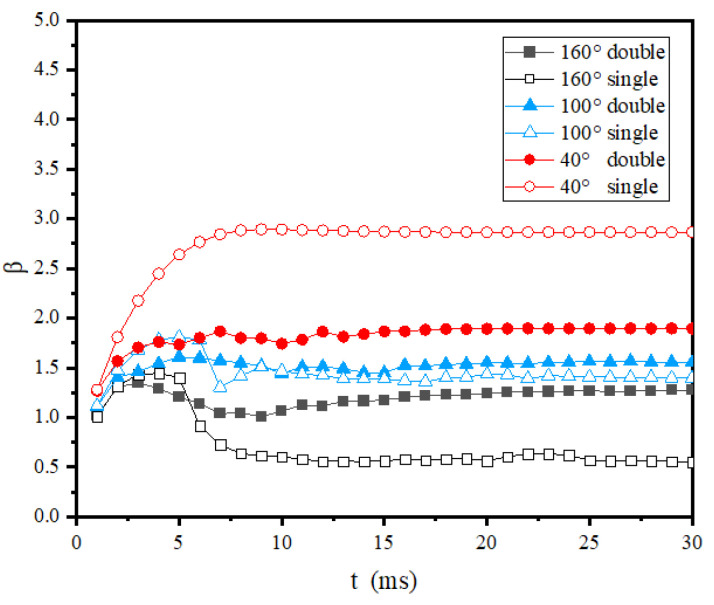
Wetting area factor evolution of supercooled double droplets impacting cold surfaces with different wettability.

**Figure 11 entropy-24-01650-f011:**
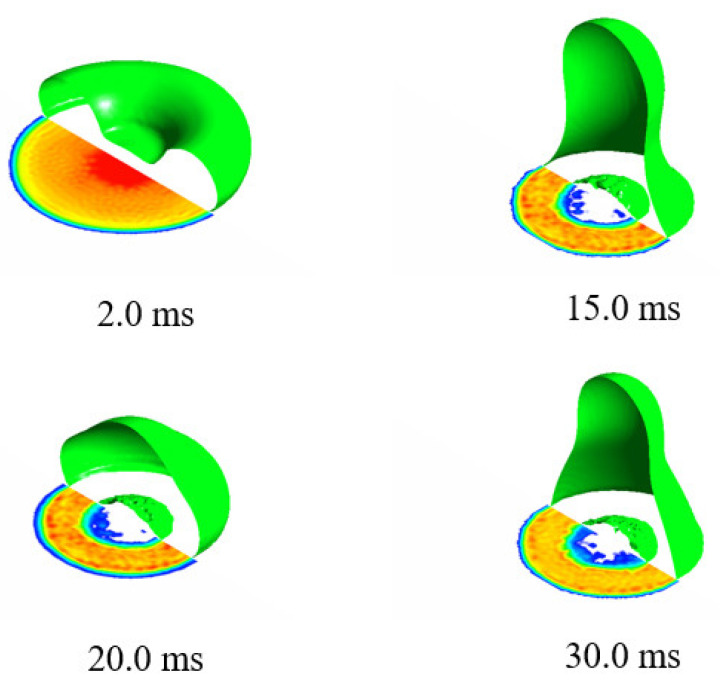
Wetting area morphological evolution of a droplet impacting a cold superhydrophobic surface ($T_{a}$ = −5 °C, $T_{s}$ = −30 °C).

**Figure 12 entropy-24-01650-f012:**
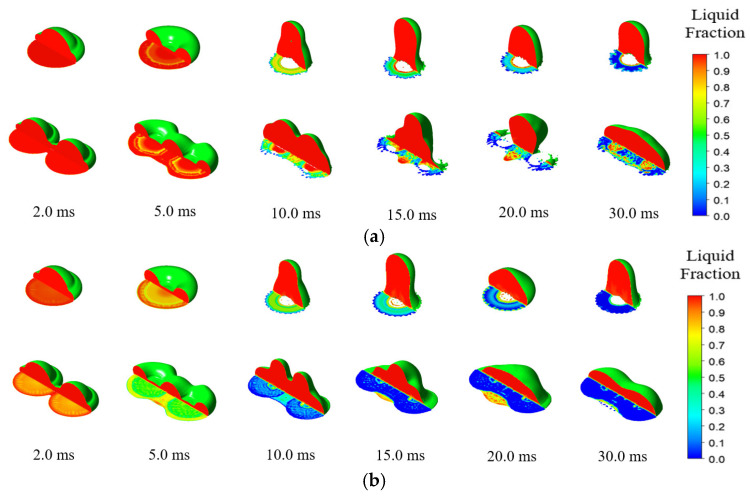
Evolution of the liquid fraction of droplets impinging on a cold surface: (**a**) $T_{a}\;$ = 15 °C, $T_{0}$ = 15 °C, $T_{s}$ = −30 °C; (**b**) $T_{a}$ = −5 °C, $T_{0\;}$ = 0.1 °C, and $T_{s}\;$ = −30 °C.

**Figure 13 entropy-24-01650-f013:**
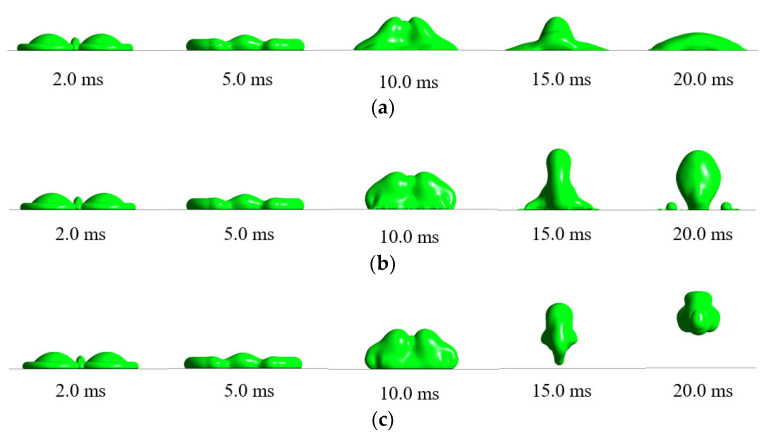
Morphology evolutions of the double droplet during the impacting–freezing process on the cold surfaces with different temperature ($V_{0}\;$ = −0.75 m/s): (**a**) full adhesion ($T_{s}$ = −30 °C); (**b**) adhesively avulsion ($T_{s}$ = −20 °C); and (**c**) full rebound ($T_{s}$ = −10 °C).

**Figure 14 entropy-24-01650-f014:**
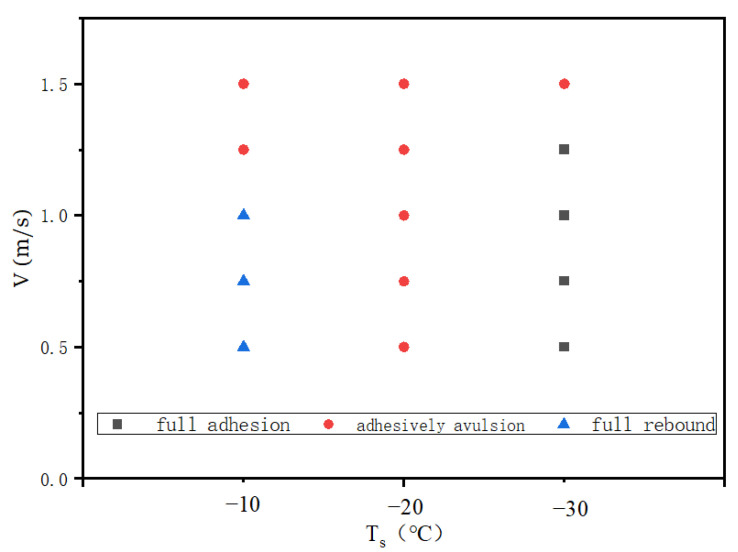
Simulation results of double droplets impacting the superhydrophobic cold surface at different surface temperatures.

**Table 1 entropy-24-01650-t001:** Simulation parameter combinations in the present work.

Case Number	Droplet Velocity (m/s)	Θ_s_ (°)	Wettability of the Surface	T_0_/T_a_/T_s_ (°C)
1	0.5	160	Super Hydrophobic	15/15/15
2				15/15/−30
3				0.1/−5/−30
4	0.5	100	Hydrophobic	15/15/15
5				15/15/−30
6				0.1/−5/−30
7	0.5	40	Hydrophilic	15/15/15
8				15/15/−30
9				0.1/−5/−30

**Table 2 entropy-24-01650-t002:** Recovery coefficient of a rebounding droplet on a superhydrophobic surface at room temperature.

	Double Droplet	Single Droplet
Recovery coefficient	0.634	0.4094

**Table 3 entropy-24-01650-t003:** Simulation conditions of droplets impact superhydrophobic cold surfaces with different velocities.

Droplet Diameter (mm)	Droplet Velocity (m/s)	Air Temperature (°C)	Surface Temperature (°C)	Droplet Temperature (°C)
2.5	0.5, 0.75, 1, 1.25, 1.5	−5	−10/−20/−30	0.1

## Data Availability

The data that support the findings of this study are available from the corresponding author upon reasonable request.

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
