# Peer review of "3D Simulations of Freezing Characteristics of Double-Droplet Impact on Cold Surfaces with Different Wettability"

_entropy, 2022, doi:10.3390/e24111650_

Round 1
Reviewer 1 Report
This manuscript numerically studied the freezing characteristics of double-droplet impact on cold surfaces by coupling the solidification and melting VOF models. The influences of surface wettability and temperature were discussed. Some useful results regarding to the droplet motion and wetting area were obtained. The manuscript could be accepted for publication after the following issues are addressed.
1. Abstract, “In the case of the hydrophilic surface, the wetting area is reduced in double-droplet impact compared to single-droplet impact; while in the case of the superhydrophobic surface, the wetting area is increased in double-droplet impact compared to single-droplet impact” could be shortened. Please check “double-droplet double-droplet” and “behaves different”.
2. P3, “The freezing of a supercooled droplet on a cold surface consists of two processes nucleation and recalescence. As the recalescence phase is fast and has a complex triggering mechanism, in this work, this stage is neglected in the simulation”. From Table 1, the droplet temperature is 0.1°C. Did the authors used supercooled droplets in their simulations? If not, this sentence could be deleted.
3. Did the authors check mesh independence for the results in Figure 1?
4. Please give the definitions of D/D0 used in Figure 1(c) and Figure 3 using a schematic diagram.
5. It’s better to use “t” rather than “time” for the abscissae Figures 1(c), 3, 7, 8, and 9.
Reviewer 2 Report
* The discussion seems inadequate, and this is too short for a reputable international journal. Many graphical presentations are similar; it is worth thinking about expressing with other graphics (if possible).
* According to the instructions of the journal: the important contents in the abstract should be well-described, such as the purpose of the research, the principal results, and the conclusions. In addition, the authors need to explicitly describe the research objectives to make them easier for readers.
* The authors should emphasize and explain the current study's novelty, which differs remarkably from previous research.
* The conclusion must answer whether the proposed method can solve the research problem and achieve the objective. How can the numerical approach answer the existing issues? What is the most important result? What are the implications for science and technology development?
* The following latest studies are very relevant to present article. The authors must read and provide complete information on this topic through including these Source:
https://doi.org/10.1016/j.solmat.2022.111786,https://doi.org/10.1016/j.est.2022.104954
* The writing of some references needs to be rechecked for accuracy
* Captions for figures and tables should be checked again. There is a significant lack of information. Please provide readers enough information on them.
Reviewer 3 Report
In this work, the freezing characteristics of double-droplet impact on three typical wettability surfaces were investigated by coupling the solidification and melting VOF models. Different temperature conditions were adopted to study the influence of icing speed on droplet behavior. Overall, the physics and results are interesting and the work shows strong merit for publication. The followings are some suggestion to improve.
1.) Several temperatures were selected and studied in this paper. What is the basis of selection?
2.) Several wettability surfaces are selected and studied in this paper. What is the basis of selection?
3.) Existing studies on phase change heat transfer. Besides the already cited papers, I recommend reviewing and citing, if suitable for your manuscript, the following articles on phase change heat transfer.
Interfacial characteristics of steam jet condensation in subcooled water pipe flow - an experimental and numerical study. Chemical Engineering Science, 2022, 251: 117457.
Concurrent Droplet Coalescence and Solidification on Surfaces With Various Wettabilities. JOURNAL OF FLUIDS ENGINEERING-TRANSACTIONS OF THE ASME, 137 (7)
Flow characteristic of steam jet condensed into a water pipe flow - a numerical study. Applied Thermal Engineering, 2022, 205: 118034.
Water droplet freezing on cold surfaces with distinct wetabilities. HEAT AND MASS TRANSFER 57 (5), pp.1-10
4.) Why the enthalpy-porosity phase change model is adopted to simulate the solidification-melting phase change process inside the droplet?
